# Updating the National Antigen Bank in Korea: Protective Efficacy of Synthetic Vaccine Candidates against H5Nx Highly Pathogenic Avian Influenza Viruses Belonging to Clades 2.3.2.1 and 2.3.4.4

**DOI:** 10.3390/vaccines10111860

**Published:** 2022-11-03

**Authors:** Yong-Myung Kang, Hyun-Kyu Cho, Sung-Jun An, Hyun-Jun Kim, Youn-Jeong Lee, Hyun-Mi Kang

**Affiliations:** Animal and Plant Quarantine Agency, 177 Hyeoksin 8-ro, Gimcheon-si 39660, Korea

**Keywords:** high pathogenic avian influenza, synthesis of HA, vaccine candidates, antigen bank, SPF chicken

## Abstract

Since 2018, Korea has been building an avian influenza (AI) national antigen bank for emergency preparedness; this antigen bank is updated every 2 years. To update the vaccine strains in the antigen bank, we used reverse genetics technology to develop two vaccine candidates against avian influenza strains belonging to clades 2.3.2.1d and 2.3.4.4h, and then evaluated their immunogenicity and protective efficacy in SPF chickens challenged with H5 viruses. The two vaccine candidates, named rgCA2/2.3.2.1d and rgES3/2.3.4.4h, were highly immunogenic, with hemagglutination inhibition (HI) titers of 8.2–9.3 log_2_ against the vaccine strain, and 7.1–7.3 log_2_ against the lethal challenge viruses (in which the HA genes shared 97% and 95.4% homology with that of rgCA2/2.3.2.1d and rgES3/2.3.4.4h, respectively). A full dose of each vaccine candidate provided 100% protection against the challenge viruses, with a reduction in clinical symptoms and virus shedding. A 1/10 dose provided similar levels of protection, whereas a 1/100 dose resulted in mortality and virus shedding by 7 dpi. Moreover, immunity induced by the two vaccines was long lasting, with HI titers of >7 log_2_ against the vaccine strain remaining after 6 months. Thus, the two vaccine candidates show protective efficacy and can be used to update the AI national antigen bank.

## 1. Introduction

H5 highly pathogenic avian influenza (HPAI) virus clades 2.3.2.1 and 2.3.4.4 continue to spread worldwide, causing great concern [1,2]. H5Nx clade 2.3.4.4 viruses have been reported in Asia, Europe, Africa, and North America [3]. Moreover, clade 2.3.2.1 H5N1 viruses have been detected in Asian countries, including China, Vietnam, and Indonesia, since 2013 [4]. In Korea, H5N1 HPAI first emerged on a poultry farm in 2003, resulting in high avian mortality; the last outbreak occurred in April 2022 [5,6]. Outbreaks of the H5N1 virus belonging to clade 2.3.2.1 occurred in 2010/2011, and outbreaks of the H5N1, H5N6, and H5N8 viruses belonging to clade 2.3.4.4 b, c, and e have occurred since 2014 [7,8,9,10,11,12,13,14]. Clade 2.3.2.1d or 2.3.4.4h viruses have not been detected in Korea; however, outbreaks have continued to occur in neighboring countries, meaning that there is the potential for introduction into Korea at any time [1,2].

Since 2018, Korea has been building an AI national antigen bank for emergency preparedness [15]; this antigen bank is updated every 2 years. The ideal AI vaccine provides the best protection against viruses that are antigenically close to field viruses [16]. Therefore, vaccines stocks need to be updated continuously to prevent infections by novel HPAI viruses [17]. Five types of vaccine strains in the AI national antigen bank were stocked and discarded from 2018 to 2020 [15]. Clades 2.3.2.1d and 2.3.4.4b have been stocked since 2020, and clades 2.3.4.4c and 2.3.4.4h have been stocked since 2021. The Korean government holds a vaccine committee meeting every year to monitor and select the epidemic strains used to update the vaccine candidate strains held in the national antigen bank. Here, we developed vaccine candidates based on clades not yet detected in Korea; to do this, we used reverse genetics approaches (including the synthesis of the HA gene).

To generate influenza vaccine candidates using reverse genetics approaches, the highly pathogenic HA gene from an original HPAI virus is either replaced with a lowly pathogenic HA gene from another virus [15], or by a synthesized lowly pathogenic HA gene [18]. The vaccine development procedure is simplified considerably by synthesizing the required genes directly, resulting in the design of a more effective vaccine [19]. This synthetic platform is an effective tool for developing vaccines against newly evolving viruses in real time [20]; it is particularly suitable for synthesizing HA genes, especially those of the most recent strains that have a high potential for introduction into Korea.

The objective of this study was to develop effective vaccine candidates to update the AI national antigen bank, and to evaluate their immunogenicity and protective efficacy against challenge viruses.

## 2. Materials and Methods

### 2.1. Viruses

Two H5 HPAI viruses were used as challenge strains and as donors of the NA gene. These strains were selected by the avian influenza vaccine committee of Animal and Plant Quarantine Agency: A/duck/Korea/Cheonan/2010(H5N1), referred to hereafter as CA/2.3.2.1, and A/duck/Korea/ES2/2016(H5N6), referred to hereafter as ES2/2.3.4.4, were isolated from a poultry farm in Korea. The viruses were propagated for 60 h in 10-day-old embryonated specific pathogen-free (SPF) eggs.

### 2.2. Synthesis of the HA Gene and Vaccine Development

The GISAID and IRD databases were used to obtain HA genome sequences, including those for clades 2.3.2.1 and 2.3.4.4. The CLC workbench program was used for nucleotide sequence translation and amino acid comparison [18]. Each HA sequence was compared, and the consensus nucleotide sequences of modified HA genes with deletion of the polybasic amino acid region were synthesized (Bioneer, Seoul, Korea). The two HA nucleotide sequences were cloned into the viral expression vector v2pHW [21], which was kindly provided by the Korea Centers for Disease Control & Prevention (KCDC), along with the NA genes from CA/2.3.2.1 and ES2/2.3.4.4, and six internal genes derived from the A/Puerto Rico/8/34 (H1N1) virus strain [22]. The two recombinant vaccine candidates (named after the candidate virus strain prefixed by ‘rg’) generated from the v2pHW constructs were transfected into 293T cells and incubated in the allantoic fluid of 10-day-old SPF chicken eggs for 2–3 days at 37 °C. Allantoic fluid was harvested, and virus growth was measured in a hemagglutination activity (HA) assay. Allantoic fluid was sub-cultured and adapted until the greatest proliferative activity was obtained, after which the 50% egg infectious dose (EID_50_) was measured, as described previously [23]. The genome sequences of the H5Nx reassortants were confirmed by sequencing. To confirm that they were LPAI, 10 chickens were challenged intranasally with 10^6^ EID_50_ per 0.1 mL of each recombinant vaccine candidate and monitored up until 14 days post-infection (dpi). Next, each recombinant vaccine candidate was inactivated in 0.1% formalin and adjusted to provide 512 HA units as a single, 51.2 HA units as a 1/10, and 5.12 HA units as a 1/100 dose when mixed (70:30) with Montanide ISA VG70 oil adjuvant (SEPPIC, La Garenne-Colombes, France) [24].

### 2.3. Phylogenetic Analysis

Phylogenetic trees were constructed using the distance-based neighbor-joining method within the MEGA (version 6.0) software package. Bootstrap analysis with 1000 replicates was used to assess the reliability of the trees [25].

### 2.4. Vaccination and Challenge of Chickens to Determine Vaccine Potency and Efficacy

To evaluate the potency (PD_50_; the dose of vaccine that protects 50% of chickens against virus challenge) and efficacy of the inactivated vaccines, 40 6-week-old SPF chickens per vaccine were divided into four groups (10 chickens per group): three vaccinated groups and one non-vaccinated (sham) group. Each bird was injected intramuscularly with a full dose, a 1/10 dose, or a 1/100 dose of vaccine mixed with Montanide ISA VG70. The sham group was inoculated with phosphate-buffered saline and ISA VG70. At 3 weeks post-vaccination (wpv), chickens were challenged intranasally with 10^6^ EID_50_ per 0.1 mL of virus (4.4 and 3.2 LD_50_ of CA/2.3.2.1 and ES2/2.3.4.4, respectively) and monitored daily for clinical signs and mortality. The PD_50_ was calculated as described previously [24], using mortality as the end point. All experiments with live H5 virus were performed in biosafety level 3 facilities and in accordance with guidelines approved by the Animal Ethics Committee of the Animal and Plant Quarantine Agency, Korea (Approval number: 2019-460).

#### 2.4.1. Serology and Antibody Assays

Blood samples were collected from all chickens before vaccination, and weekly thereafter. Blood samples were also obtained from all living chickens at 14 days post-challenge (dpc). Hemagglutination inhibition (HI) assays were performed using standard methods [26] against vaccine strain antigens and inactivated challenge antigens.

#### 2.4.2. Post-Challenge Virus Shedding

The protective efficacy of each vaccine was determined by evaluating clinical signs, mortality, and virus shedding after intranasal challenge. Oropharyngeal (OP) and cloacal (CL) swabs were collected from animals, including those that died, at 1, 3, 5, 7, 10, and 14 dpi. Each OP or CL sample was suspended in 1 mL of maintenance medium containing antibiotic–antimycotic mixture (Invitrogen, Carlsbad, CA, USA). Samples were used to inoculate cultures of dermal fibroblast 1 cells, and virus growth was determined by detection of cytopathic effects and HA activity. Virus titers were calculated, as described elsewhere [24], and the limit of virus detection was <1. The statistical significance of differences between measurements was determined using Student’s *t*-test, with a *p*-value < 0.05 indicating a significant difference.

### 2.5. Antibody Persistence

To determine the longevity of protection afforded by the inactivated H5 vaccines, 10 chickens (6 weeks old) were vaccinated with a full dose. Blood samples were collected monthly post-vaccination (continuing for 6 months). HI assays against vaccine strain antigens were performed using standard methods [26].

### 2.6. Statistical Analysis

Data were analyzed using Prism version 5.0 software (GraphPad Software, La Jolla, CA, USA). Comparison of serum titers between groups was made by one-way analysis of variance. Survival rates among groups were analyzed using the log-rank test. A *p* < 0.05 value was considered statistically significant.

## 3. Results

### 3.1. Vaccine Development

Two H5 AI vaccines were established using reverse genetics techniques. After the reassortant vaccine candidate strains were rescued successfully by transfection into 293T cells, the viruses were adapted and propagated by passage in SPF chicken eggs. The EID_50_ values of the final allantoic fluids were 7.8–8.7 log_10_ per 0.1 mL, and the HA titers were >8 log_2_ (Table 1). The monitoring of the SPF chickens for 14 dpi with the reassortant vaccine candidates revealed no clinical signs or mortality, confirming that the viruses were LPAI (data not shown). A genetic analysis of the HA sequence revealed that the vaccine candidates belonged to clades 2.3.2.1d and 2.3.4.4h (Figure 1). Homology between the new vaccine strains and the challenge strain was 97.0% (clade 2.3.2.1d) and 95.4% (clade 2.3.4.4h), and the mutations related to vaccine protection were I to T at amino acid position 71 (rgCA2/2.3.2.1d vs. CA/2.3.2.1), and S to A at amino acid position 133 (rgES3/2.3.4.4h vs. ES2/2.3.4.4) (Table 2).

### 3.2. Vaccine Potency

#### 3.2.1. Clinical Protection

Vaccination with a full or 1/10 dose of rgCA2/2.3.2.1d or rgES3/2.3.4.4h conferred 100% clinical protection, with no clinical signs after challenge with CA/2.3.2.1 and ES2/2.3.4.4 (Figure 2). However, vaccination with a 1/100 dose of each strain resulted in some mortality and clinical signs of infection. Vaccination with rgCA2/2.3.2.1d at a 1/100 dose resulted in 80% mortality by 6 dpi (Figure 2A); the two chickens that died on day 6 showed neurological signs. With respect to rgES3/2.3.4.4h, vaccination with a 1/100 dose resulted in 40% mortality on day 4 (Figure 2B), with the four chickens that died showing neurological signs and diarrhea. The mean time to death in the 1/100 dose vaccination groups was 2.7–4.5 days (Table 1). For sham-treated chickens, the mean time to death was 2.2–2.6 days. Clinical efficacy was also indicated by the vaccine potency results: rgCA2/2.3.2.1d and rgES3/2.3.4.4h showed high potency values of 42 and 68, respectively (Table 1).

#### 3.2.2. Serology

Following vaccination, antibody titers increased both pre- and post-challenge (Figure 3). Vaccination with a full and 1/10 dose resulted in seroconversion prior to viral challenge at 21 dpv, with mean HI titers of 7.1–7.3 and 4.0–5.1 log_2_, respectively, against challenge strain antigens compared with 8.2–9.3 log_2_ and 4.4–5.9 log_2_, respectively, against vaccine strain antigens (Table 1). By 14 dpi, the mean HI titers were 9.0–9.7 log_2_ and 7.7–7.8 log_2_ for the full and 1/10 doses, respectively. Vaccination with a 1/100 dose produced measurable responses at 21 dpv in some (but not all) chickens, with HI titers of 0.5 against the challenge strain antigen (compared with 0.4–0.5 log_2_ against the vaccine strain antigen). Following viral challenge, all surviving chickens had measurable HI titers (7.7–7.8 log_2_) at 14 dpi. None of the sham-vaccinated chickens had detectable HI titers before viral challenge (Figure 3).

#### 3.2.3. Virus Shedding

Virus shedding was observed in the rgCA2/2.3.2.1d and rgES3/2.3.4.4h vaccinated groups from 1–7 dpi and after viral challenge (Figure 4). After vaccination with a full dose of rgCA2/2.3.2.1d and rgES3/2.3.4.4h, the shedding of the virus (mainly in the OP) was observed from 1–7 dpi, with viral titers of 10^1.5^–10^2.0^ TCID_50_ (50% tissue culture infectious dose) per 0.1 mL. At 1 and 3 dpi, virus titers from chickens vaccinated with a full dose of rgCA2/2.3.2.1d, and those at 1 dpi from chickens vaccinated with a full dose of rgES3/2.3.4.4h, were significantly lower than those in sham-vaccinated chickens (*p* < 0.05) (Figure 4). After vaccination with a 1/10 dose of rgCA2/2.3.2.1d and rgES3/2.3.4.4h, virus shedding was detected from 1–7 dpi, with viral titers of 10^1.3^ to 10^3.3^ TCID_50_ per 0.1 mL. Vaccinations with a 1/100 dose of rgCA2/2.3.2.1d and rgES3/2.3.4.4h resulted in virus shedding by both surviving and dead chickens, with titers of 10^1.7^–10^7.3^ TCID_50_ per 0.1 mL detected in OP swab samples from 1 to 5 dpi, and titers of 10^1.7^–10^6.0^ TCID_50_ per 0.1 mL detected in CL swab samples. In the sham group, virus shedding was detected in both surviving and deceased chickens, with mean titers of 10^2.5^ to 10^6.1^ TCID_50_ per 0.1 mL in both OP and CL swabs.

### 3.3. Antibody Persistence

At 3 wpv, chickens inoculated with a full dose of the two inactivated vaccines had mean HI titers against the vaccine strain antigen of 10.3 log_2_ (rgCA2/2.3.2.1d) and 9.9 log_2_ (rgES3/2.3.4.4h). The mean HI titers peaked between 3 and 8 wpv, and decreased slightly up to 24 wpv, while remaining above 7 log_2_(8.3 and 7.8 log_2_ at 24 wpv) (Figure 5).

## 4. Discussion

Korea has been building an AI national antigen bank since 2018; this bank is updated every 2 years. Two viruses within clades 2.3.2.1d and 2.3.4.4h, which have not yet been introduced into Korea but have emerged in neighboring countries, were selected as vaccine candidates for updating the AI national antigen bank. The two vaccine strains were developed using a reverse genetics approach to synthesize the HA gene; this was combined with internal genes from the PR8 virus to produce the vaccine candidates. Here, we evaluated the immunogenicity and protective efficacy of these vaccine candidates in SPF chickens challenged with viruses from the same clades.

Successful AI vaccination reduces clinical signs and death, thereby preventing economic loss; in addition, vaccinated birds shed less viruses, thereby reducing transmission [27]. It was reported that vaccination with the H5/H7 bivalent vaccine reduced the prevalence of H7N9 outbreaks in poultry [28]. Here, we show that a full dose of the candidate vaccines rgES3/2.3.4.4h and rgCA2/2.3.2.1d generated a HI titer of >7 and provided 100% clinical protection against challenge at both a full and 1/10 dose, although they were challenged with different viruses within the same clade since there were no parent viruses having 100% of HA homology. Furthermore, chickens vaccinated with a full or 1/10 dose shed much less virus than sham-vaccinated birds or birds receiving a 1/100 dose (*p* < 0.05) (Figure 4). Birds receiving a full dose of rgES3/2.3.4.4h or rgCA2/2.3.2.1d showed HI titers of 8.2 and 9.3 log_2_, respectively, against the vaccine strain antigen, which is 2–4 times higher than that against the challenge strain antigen (7.1 and 7.3 log_2_, respectively). A previous study reported that 100% homology between the vaccine and challenge strains resulted in 100% protection, a reduction in clinical signs, a significant reduction in virus shedding, and a HI titer of ≥8 log_2_ [12]. Here, we found that the high HI antibody titers against the vaccination strain antigen persisted (>7 log_2_) for 6 months in the group that received a full dose. The WOAH criteria suggest that a vaccine should meet minimum antigen requirements: a PD_50_ of 50 or 3 μg of hemagglutinin per dose. The minimum HI serological titers in field birds should be 1/32 to protect against mortality, or greater than 1/128 to provide a reduction in the replication and shedding of the challenge virus [29]. Thus, the two vaccine candidates developed herein satisfy with WOAH criteria, despite the fact that the homology between the challenge and vaccine strains is less than 100%. The Re-8 vaccine developed in China also showed lower protective efficacy against clade 2.3.2.1b with homology less than 100%, but higher protective efficacy against clade 2.3.4.4 [30]. Taken together, the data suggest that the two vaccine candidates have high immunogenicity and protective efficacy. We observed virus shedding in all vaccinated groups, particularly up until 7 dpi in the group vaccinated with a 1/100 dose of rgCA2/2.3.2.1d, and the groups vaccinated with a full or 1/10 dose of rgES2/2.3.4.4h, with viral titers of 1.5–7.7 TCID_50_/0.1 mL (Figure 4). A previous study showed that, assuming comparable host immune responses, maximizing the antigenic similarity of the HA protein between the vaccine and challenge viruses results in the best protection against mortality and virus shedding [27].

Another study showed that a peptide in the main epitope, LVLWHIHHP, bound to two mAbs and was located in the HA head region [31]. In the present study, the vaccine strain rgCA2/2.3.2.1d differs from the CA challenge virus with respect to one amino acid; this amino acid resides within the LVLWHIHHP peptide. In addition, mutations at amino acid positions 69, 71, 83, 95, 133, 140, 162, 183, 189, 194, and 270 (H5 numbering) (Table 2) of HA may result in the loss of vaccine protection [32]. Here, we confirmed an I to T substitution at amino acid position 71 in rgCA2/2.3.2.1d (compared with CA/2.3.2.1), and an S to A substitution at position 133 in rgES3/2.3.4.4h (compared with ES2/2.3.4.4) (Table 2). A previous study reported that differences in virulence between challenge viruses lead to differences in protective efficacy [33]. Thus, not only the degree of homology between the vaccine strain and the field strain, but also occurrence of mutations in the major epitope, affect vaccine efficacy. In the case of synthesis of the HA gene, the vaccine and field strains can be considered homologous and to share the main epitope.

In conclusion, we generated two inactivated vaccines using a reverse genetics approach. Both vaccine candidates showed good protective efficacy (no mortality and reduced virus shedding) against the challenge viruses when used at a full dose. These results suggest that both vaccine candidates developed can be used to update the vaccine strains in the AI national antigen bank.

## Figures and Tables

**Figure 1 vaccines-10-01860-f001:**
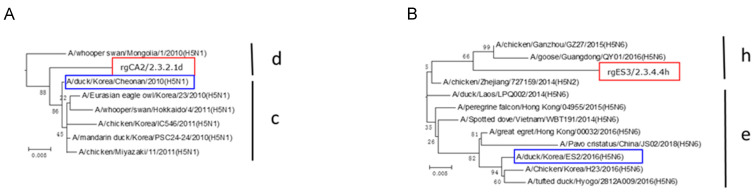
Phylogenetic trees of clades 2.3.2.1 and 2.3.4.4 (HA; H5). The vaccine strains developed by reverse genetics are indicated in red. Each tree was built using the distance-based neighbor-joining method in MEGA6 software (bootstrap value = 1000). Red boxes indicate challenge strain and blue boxes indicate vaccine strain. (**A**) rgCA2/2.3.2.1d. (**B**) rgES3/2.3.4.4h.

**Figure 2 vaccines-10-01860-f002:**
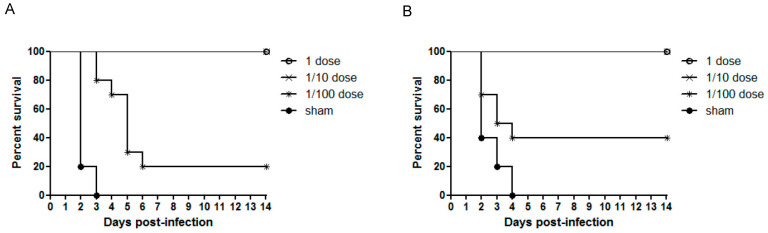
Survival of chickens inoculated with a single dose of one of the two inactivated vaccines (a full dose, a 1/10 dose, a 1/100 dose, or sham) and then challenged with highly pathogenic H5 viruses. (**A**) rgCA2/2.3.2.1d. (**B**) rgES3/2.3.4.4h.

**Figure 3 vaccines-10-01860-f003:**
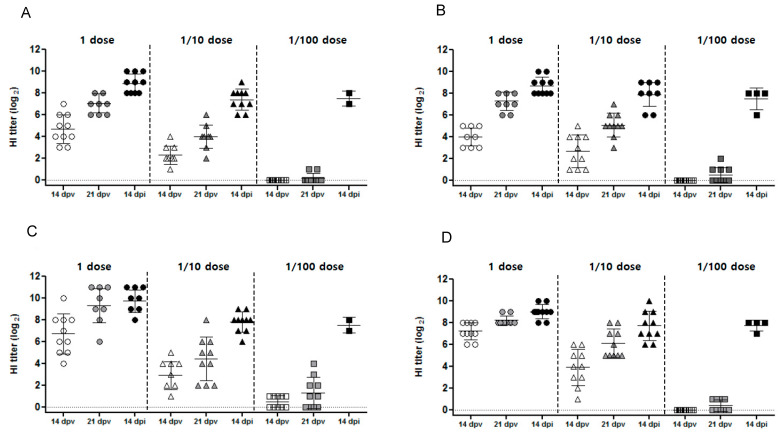
Hemagglutination inhibition (HI) assay titers in chickens vaccinated at different times with a full dose, a 1/10 dose, or a 1/100 dose of each candidate vaccine. HI titers were assessed at 14 days post-vaccination (dpv), at 21 dpv, and at 14 days post-infection. (**A**) HI titer of rgCA2/2.3.2.1d against challenge virus antigen; (**B**) HI titer of rgES3/2.3.4.4h against challenge virus antigen; (**C**) HI titer of rgCA2/2.3.2.1d against vaccine strain antigen; and (**D**) HI titer of rgES3/2.3.4.4h against vaccine strain antigen. Individual data points are shown, along with the mean and standard error.

**Figure 4 vaccines-10-01860-f004:**
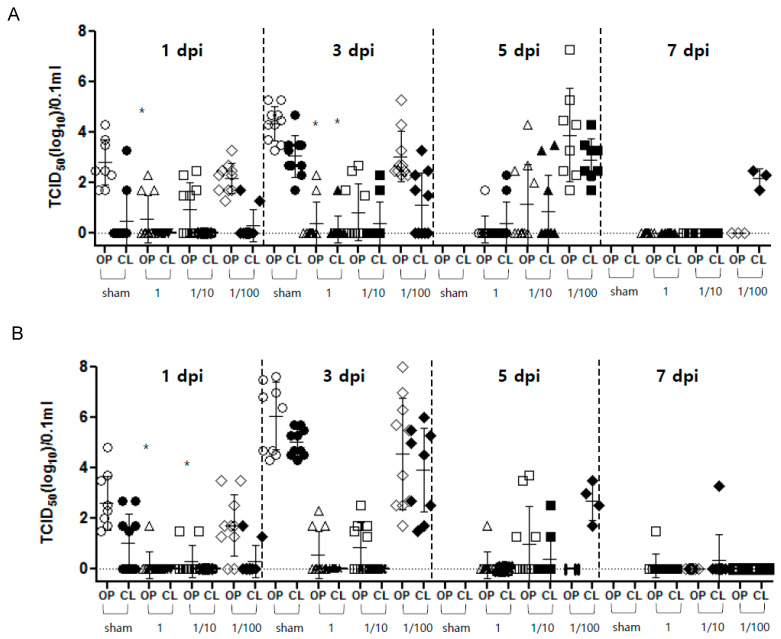
Shedding of virus in oropharyngeal and cloacal swabs taken from chickens inoculated with inactivated vaccines (full dose, 1/10 dose, 1/100 dose, or sham) was assessed at 1, 3, 5, and 7 days post-infection with highly pathogenic H5 viruses (Figure 4). Viral titers are expressed as log_10_TCID_50_ (50% tissue culture infectious dose) per 0.1 mL, with error bars. (**A**) rgCA2/2.3.2.1d. (**B**) rgES3/2.3.4.4h. The lower limit of detection was 0.3 log_10_ TCID_50_ per 0.1 mL. * *p* value < 0.05.

**Figure 5 vaccines-10-01860-f005:**
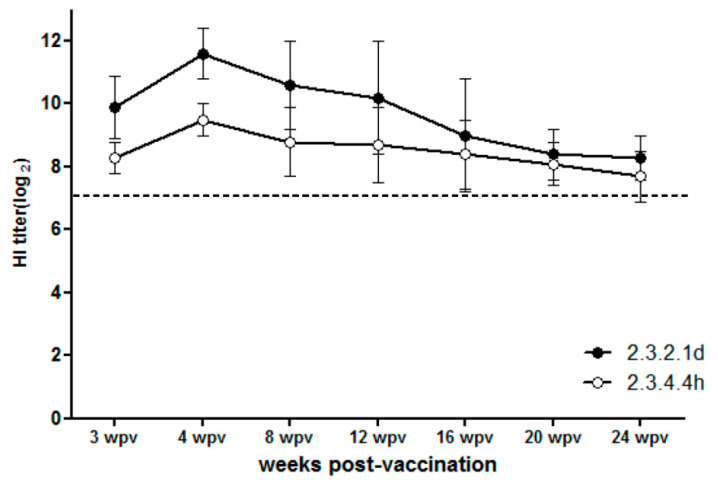
Antibody persistence following vaccination with a full dose of each candidate vaccine. Hemagglutination inhibition (HI) assay titers were measured during the 24 weeks post-vaccination. Titers are expressed as log_2_ values. The horizontal dotted line indicates a HI titer of 7 log_2_, which is the standard threshold for prevention of virus shedding.

**Table 1 vaccines-10-01860-t001:** Data from specific pathogen-free chickens vaccinated with different doses of inactivated vaccines against highly pathogenic avian influenza virus.

Vaccine	Growth Properties in Eggs		Challenge Virus	Vaccine Efficacy								
	EID_50_ (log_10_/0.1 mL)	HA Titer (log_2_)	Homology with the Vaccine (%)	Antigen Dose ^1^	HI Titer (log_2_) ^4^ Against		Clinical Signs	Peak Shedding (3 dpi) ^3^		Survival (%) (MDT) ^2^	PD_50_	Antibody Persistence (24 wpv) ^5^
					Challenge Strain Antigen	Vaccine Strain Antigen		OP	CL			
rgCA2/2.3.2.1d	8.2	8	CA/2.3.2.1 97.0	1	10/10 (7.1)	10/10 (9.3)	Diarrhea	2/10 * (2.0)	1/10 * (1.7)	100	42	8.3
				1/10	10/10 (4.0)	10/10 (4.4)	Diarrhea	4/10 (2.1)	2/10 (2.0)	100		
				1/100	2/10 (0.5)	5/10 (0.5)	Lethargy, green feces, anorexia, Death	10/10 (3.1)	5/10 (2.3)	20 (4.5)		
				Sham	0/10	0/10	Death	10/10 (4.4)	10/10 (3.1)	0 (2.2)		
rgES3/2.3.4.4h	8.4	8	ES2/2.3.4.4 95.4	1	10/10 (7.3)	10/10 (8.2)	Diarrhea	3/10 (1.9)	0/10	100	68	7.8
				1/10	10/10 (5.1)	10/10 (5.9)	Diarrhea	5/10 (1.7)	0/10	100		
				1/100	4/10 (0.5)	4/10 (0.4)	Lethargy, green feces, Death	10/10 (4.6)	10/10 (3.9)	40 (2.7)		
				Sham	0/10	0/10	Death	10/10 (6.1)	10/10 (5.0)	0 (2.6)		

^1^ A full dose (1) contained 512 HAU (hemagglutination units). ^2^ MDT = mean death time (days). ^3^ Number of virus-positive/total birds in the group (mean shed virus titer). ^4^ Number of serology-positive/total number surviving birds in the group (mean HI titer). ^5^ Mean HI titer at 24 weeks post-vaccination (wpv) with a single full-strength dose. * *p*-value < 0.05. Abbreviations: EID_50_—50% egg infectious dose; HA—hemagglutination activity; NT—not tested; dpi—days post-infection; OP—oropharyngeal; CL—cloacal; HI—hemagglutination inhibition; and PD_50_—50% protective dose.

**Table 2 vaccines-10-01860-t002:** Comparison of variable amino acid sequences related to vaccine protection.

Variable Residue ^1^	2.3.2.1		2.3.4.4	
	Vaccine Strain ^2^	Challenge Strain	Vaccine Strain ^2^	Challenge Strain
	rgCA2/2.3.2.1d	CA/2.3.2.1	rgES3/2.3.4.4h	ES2/2.3.4.4
69	E	E	E	E
71	T	I	I	I
83	A	A	L	L
95	F	F	A	A
133	S	S	A	S
140	N	N	V	V
162	I	I	I	I
183	D	D	A	A
189	R	R	L	L
194	P	P	T	T
270	E	E	E	E
Homology ^3^ (%)	97.0		95	

^1^ H5 numbering, including the signal peptide. ^2^ Reverse genetics vaccine strain, including the synthetic consensus H5 HA sequence. ^3^ Homology between vaccine and challenge strains barring the cleavage site sequences.

## Data Availability

GISAID: https://platform.epicov.org/epi3/cfrontend (accessed on 13 April 2021); IRD: https://www.fludb.org/brc/home.spg?decorator=influenza. (accessed on 13 April 2021).

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
