# Peer review of "Updating the National Antigen Bank in Korea: Protective Efficacy of Synthetic Vaccine Candidates against H5Nx Highly Pathogenic Avian Influenza Viruses Belonging to Clades 2.3.2.1 and 2.3.4.4"

_vaccines, 2022, doi:10.3390/vaccines10111860_

Round 1

Reviewer 1 Report

Comment to viruses

In this study, Kang et al. developed two H5 vaccines by reverse genetics and evaluated their protective efficacy in chickens. These vaccine strains will contributed to prevent the invasion of the circulating H5Nx viruses in Asia. There are few questions should be addressed and I suggest the manuscript should be edited by a naive English expert or company.

Major comment

1.In table 2, the vaccine and challenge viruses were marked belong to 2.3.2.1, such as rgCA2/2.3.2.1c,CA/2.3.2.1c,but the text indicated they belong to 2.3.2.1d.

2. How to confirmed 1/10 and 1/100 dose? As described line 89, 512 HAU were a single full dose. So 51.2 HAU 5.12HAU were seemed as 1/10 and 1/100 dose, respectively? The volume of the vaccine was not referred in the manuscript.

3. Figure 1. just indicated the vaccine strain in the phylogenetic tree, but missed the challenge viruses

4. What is the meaning of ES2/2.3.4.4e in Table 1? I did not get any information about ES2/2.3.4.4e in the text.

5. In 2.1 Viruses section “A/duck/Korea/Cheonan/2010(H5N1) referred to hereafter as CA2/2.3.2.1, and A/duck/Korea/ES2/2016(H5N6), referred to hereafter ES2/2.3.4.4,” but in the table or the figure the two viruses were referred to CA2/2.3.2.1c, CA2/2.3.2.1d, ES2/2.3.4.4h

6.What is the meaning of figure 3? The figure 3 was not mentioned in the text.

7.As shown in figure 4, the two vaccines can not prevent the viral shedding after vaccination by 1 full dose at 1, 3, 5, or 7 dpi..

8. The discussion lacks the discussion of the two vaccines in this study with the H5/H7 used in China. It seems that the H5 vaccines used in China were more protective than the two vaccines used in this study (PMID: 34757542; PMID: 30414008; PMID: 27309064; doi: 10.1016/S2095-3119(22)63904-2). Also these publications should be referred by this study.

9.Why the HA sequences of rgCA2/2.3.2.1d and rgES3/2.3.4.4h were selected as the HA gene donor of the vaccines? The two HA genedid not show high similarity with the currently circulated H5Nx viruses.

10.Several important references were missed when introduction the circulation of H5Nx virus in Asia (PMID: 35699072; PMID: 35380505; PMID: 36093829; PMID: 3544615; PMID: 35413157).

Author Response

Response to Reviewer 1 Comments

Comment to viruses

In this study, Kang et al. developed two H5 vaccines by reverse genetics and evaluated their protective efficacy in chickens. These vaccine strains will contributed to prevent the invasion of the circulating H5Nx viruses in Asia. There are few questions should be addressed and I suggest the manuscript should be edited by a naive English expert or company.

- Response :  Thank you for your comment. I have already edited by English expert before submission, but I have checked again by native English expert as your suggestion.

Major comment

  1. In table 2, the vaccine and challenge viruses were marked belong to 2.3.2.1, such as rgCA2/2.3.2.1c,CA/2.3.2.1c,but the text indicated they belong to 2.3.2.1d.

 - Response : I agree with your comment. First of all, I apologize for my error. I matched the challenge virus to be CA/2.3.2.1 and ES2/2.3.4.4, and the vaccine given to rgCA/2.3.2.1d and rgES3/2.3.4.4h, respectively, between table and manuscript. [Table 2]  

  1. How to confirmed 1/10 and 1/100 dose? As described line 89, 512 HAU were a single full dose. So 51.2 HAU 5.12HAU were seemed as 1/10 and 1/100 dose, respectively? The volume of the vaccine was not referred in the manuscript.

- Response : I agree with your comment. Vaccine antigens with 1/10 and 1/100 doses were made by serial dilution of single dose antigen with 512 HAU. We confirmed HAU by HA test. So, I refered the volume of the vaccine such as 1/10(51.21HAU) and 1/100 (5.12HAU) in the manuscript. [Line 90-91]

  1. Figure 1. just indicated the vaccine strain in the phylogenetic tree, but missed the challenge viruses

- Response : I agree with your comment. I indicated challent virus using blue boxes and added legend in Figure 1 as you recommed. [Figure 1]

  1. What is the meaning of ES2/2.3.4.4e in Table 1? I did not get any information about ES2/2.3.4.4e in the text.

- Response : I agree with your comment. I modified ES2/2.3.4.4e to ES2/2.3.4.4 as you mentioned in #1 comment together. [Table 1]

  1. In 2.1 Viruses section “A/duck/Korea/Cheonan/2010(H5N1) referred to hereafter as CA2/2.3.2.1, and A/duck/Korea/ES2/2016(H5N6), referred to hereafter ES2/2.3.4.4,” but in the table or the figure the two viruses were referred to CA2/2.3.2.1c, CA2/2.3.2.1d, ES2/2.3.4.4h

- Response : I agree with your comment. I apologize for my error. As your comment in #1 and #4 together, I matched the challenge virus to be CA/2.3.2.1 and ES2/2.3.4.4, and the vaccine given to rgCA2/2.3.2.1d and rgES3/2.3.4.4h, respectively, both table and manuscript.  

  1. What is the meaning of figure 3? The figure 3 was not mentioned in the text.

- Response : I agree with your comment. I mentioned figure 3 in manuscript. [Line 198]

  1. As shown in figure 4, the two vaccines can not prevent the viral shedding after vaccination by 1 full dose at 1, 3, 5, or 7 dpi.

- Response : Thank you for your comment. It is not matched with difference of 3~5% homology with vaccine and challenge strain, and peptide in the main epitope was mutated as mentioned in Discussion section. Due to these reasons, two vaccines can not prevent the viral shedding against challenge viruss.    

  1. The discussion lacks the discussion of the two vaccines in this study with the H5/H7 used in China. It seems that the H5 vaccines used in China were more protective than the two vaccines used in this study (PMID: 34757542; PMID: 30414008; PMID: 27309064; doi: 10.1016/S2095-3119(22)63904-2). Also these publications should be referred by this study.

- Response : Thank you for your comment. I mentioned the vaccine effect by citing of references in Discussion section as you recommend. [Line 248-249, Line 268-270]

  1. Why the HA sequences of rgCA2/2.3.2.1d and rgES3/2.3.4.4h were selected as the HA gene donor of the vaccines? The two HA genes did not show high similarity with the currently circulated H5Nx viruses.

- Response : Thank you for your comment. The purpose of AI antigen bank is to establish various clades of vaccine strain that can introduce into Korea. Clade 2.3.4.4b virus that are currently circulating world are already stockpiled in antigen bank. Therefore, we are going to establish two strains(clade 2.3.2.1d and 2.3.4.4h) that have recently occurred in neighboring countries, not occured in Korea, and not currently established as antigen banks. Furthermore, all of vaccine strains were selected by National AI vaccine committee held in annually.

  1. Several important references were missed when introduction the circulation of H5Nx virus in Asia (PMID: 35699072; PMID: 35380505; PMID: 36093829; PMID: 3544615; PMID: 35413157).
  • Response : Thank you for your comment. I added references as you recommend. [Line 36]

Reviewer 2 Report

Kang et al. conducted the evaluation of the trial vaccine for high pathogenicity avian influenza caused by H5Nx infection. To stock the effective vaccine, the authors developed the vaccine strain according to the sequence data of the isolates isolated recently in nearby. Evaluation of the vaccine candidate was conducted in the view of antigenicity, phylogenic, survival rate of birds and virus shed from birds. The strategy for vaccine development described in the current manuscript is reasonable and beneficial to the audience to know the protocol of proper vaccine development. However, the authors need several minor improvements from the current version of the manuscript to be published in Vaccines.

<Minor comments>

L 102; Please indicate the 50% of chicken lethal dose of challenge virus as well.

Table 1 and others; I am so confused that the authors used three different names; rgCA3/2.3.2.1d (Table 1),  rgCA2/2.3.2.1c (Table 2),  and rgCA2/2.3.2.1d (L 204). Please correct them.

L184-188; I cannot catch why the authors indicate the mean of HI titer with ranges. What did the range of average HI titer mean? Please clearly explain this.

L225: HI titers against two vaccine strain described in here are different from the ones in Table 1. Please explain the possible reason(s) of the data variety among these two study settings.

L231-232: This description might be somehow contradictory to the results of virus shed in Figure 4; groups of birds over than 128 HI shed infectious viruses. Please add one paragraph in Discussion part about the association between the virus shed and HI titers of vaccinated birds.

L248-249: The results of statistical analysis described in here were not described in Table 1.

Author Response

Response to Reviewer 2 Comments

Kang et al. conducted the evaluation of the trial vaccine for high pathogenicity avian influenza caused by H5Nx infection. To stock the effective vaccine, the authors developed the vaccine strain according to the sequence data of the isolates isolated recently in nearby. Evaluation of the vaccine candidate was conducted in the view of antigenicity, phylogenic, survival rate of birds and virus shed from birds. The strategy for vaccine development described in the current manuscript is reasonable and beneficial to the audience to know the protocol of proper vaccine development. However, the authors need several minor improvements from the current version of the manuscript to be published in Vaccines.

<Minor comments>

L 102; Please indicate the 50% of chicken lethal dose of challenge virus as well.

Response : Thank you for your comment. I added “4.4 and 3.2 LD50 of CA/2.3.2.1 and ES2/2.3.4.4, respectively” to express LD50 of two challenge viruses in the manuscript. [Line 105]

Table 1 and others; I am so confused that the authors used three different names; rgCA3/2.3.2.1d (Table 1),  rgCA2/2.3.2.1c (Table 2),  and rgCA2/2.3.2.1d (L 204). Please correct them.

Response : I agree with your comment. I apologize for my error. I matched the challenge virus to be CA/2.3.2.1 and ES2/2.3.4.4, and the vaccine given to rgCA2/2.3.2.1d and rgES3/2.3.4.4h, respectively, both table and manuscript [Table 1, Table 2, Line 204] as reviewer 1 mentioned same points together.

L184-188; I cannot catch why the authors indicate the mean of HI titer with ranges. What did the range of average HI titer mean? Please clearly explain this.

Response : Thank you for your comment. The range of average HI titer means range between each HI titer of two types of vaccines. For example, I wrote the range of average HI titer to 7.1 – 7.3 in case that mean HI titers are 7.1 in 3 weeks post rgCA2/2.3.2.1d vaccination and 7.3 in 3 weeks post rgES3/2.3.4.4h vaccination.

L225: HI titers against two vaccine strain described in here are different from the ones in Table 1. Please explain the possible reason(s) of the data variety among these two study settings.

Response : Thank you for your comment. Data in antibody persistence section of Table 1 are HI titer at 24 weeks post vaccination, and HI titers described in L225 of manuscript are those at 3 weeks post vaccine. Therefore, HI titers are different. However, I added HI titer at 24 weeks post vaccination to avoid confusion. [Line 231]

L231-232: This description might be somehow contradictory to the results of virus shed in Figure 4; groups of birds over than 128 HI shed infectious viruses. Please add one paragraph in Discussion part about the association between the virus shed and HI titers of vaccinated birds.

Response : Thank you for your comment. I understand that you thought contradictory results that there is virus shedding despite of HI titer more than 7 log2. It was mentioned in WOAH terrestrial Manual (Ref.25) and discussion section (Line 262-266) that “minimum HI serological titers in field birds should be 1/32 to protect from mortality or greater than 1/128 to provide reduction in challenge virus replication and shedding”. That means reduction, not prevention of virus shdding in case of 7 log2(1/128) in HI titer. Therefore, I modified “inhibit” to “provide reduction” in Discussion section for avoiding confusion. [Line 265]

L248-249: The results of statistical analysis described in here were not described in Table 1.

Response : Thank you for your comment. Virus shedding result only in 3 dpi are presented in Table 1. Therefore, “[Table 1]” in the results of statistical analysis Line 255 was modified to “(Figure 4)” and it was simultaneously added “*” and p-value in data with statistical significance. [Line 255, Table 1]

Round 2

Reviewer 1 Report

The authors have addressed my concern.